# Therapeutic Targeting of the Leukaemia Microenvironment

**DOI:** 10.3390/ijms22136888

**Published:** 2021-06-26

**Authors:** Vincent Kuek, Anastasia M. Hughes, Rishi S. Kotecha, Laurence C. Cheung

**Affiliations:** 1Leukaemia Translational Research Laboratory, Telethon Kids Cancer Centre, Telethon Kids Institute, Perth, WA 6009, Australia; vincent.Kuek@telethonkids.org.au (V.K.); Anastasia.Hughes@telethonkids.org.au (A.M.H.); Rishi.Kotecha@health.wa.gov.au (R.S.K.); 2Curtin Medical School, Curtin University, Perth, WA 6102, Australia; 3School of Biomedical Sciences, University of Western Australia, Perth, WA 6009, Australia; 4Department of Clinical Haematology, Oncology, Blood and Marrow Transplantation, Perth Children’s Hospital, Perth, WA 6009, Australia; 5School of Medicine, University of Western Australia, Perth, WA 6009, Australia

**Keywords:** bone marrow microenvironment, leukaemia, mesenchymal stem cell, endothelial cell, osteoclast, osteoblast, therapeutic agents

## Abstract

In recent decades, the conduct of uniform prospective clinical trials has led to improved remission rates and survival for patients with acute myeloid leukaemia and acute lymphoblastic leukaemia. However, high-risk patients continue to have inferior outcomes, where chemoresistance and relapse are common due to the survival mechanisms utilised by leukaemic cells. One such mechanism is through hijacking of the bone marrow microenvironment, where healthy haematopoietic machinery is transformed or remodelled into a hiding ground or “sanctuary” where leukaemic cells can escape chemotherapy-induced cytotoxicity. The bone marrow microenvironment, which consists of endosteal and vascular niches, can support leukaemogenesis through intercellular “crosstalk” with niche cells, including mesenchymal stem cells, endothelial cells, osteoblasts, and osteoclasts. Here, we summarise the regulatory mechanisms associated with leukaemia–bone marrow niche interaction and provide a comprehensive review of the key therapeutics that target CXCL12/CXCR4, Notch, Wnt/b-catenin, and hypoxia-related signalling pathways within the leukaemic niches and agents involved in remodelling of niche bone and vasculature. From a therapeutic perspective, targeting these cellular interactions is an exciting novel strategy for enhancing treatment efficacy, and further clinical application has significant potential to improve the outcome of patients with leukaemia.

## 1. Introduction

The bone marrow microenvironment (BMM) is a primary site for haematopoiesis and consists of unique bone marrow (BM) cells, which facilitate haematopoietic cell regeneration and renewal in specialised regions known as “niches”. These BM niches provide critical factors and signals that support the physiological maintenance of haematopoietic stem cells (HSCs) as well as the formation of lineage-committed cells in a highly regulated and controlled manner [1]. However, emerging evidence has pointed to their role in malignant transformation and, subsequently, pathogenesis of leukaemia. While genomic aberrations underpin leukaemic transformation, the BMM is being increasingly recognised as a key player in leukaemia progression, treatment resistance, and disease relapse. The ability of leukaemic stem or progenitor cells to interact with their neighbouring supportive cells can severely disrupt haematopoietic homeostasis, leading to the formation of a leukaemic-friendly niche or “safe haven” for malignant cells to escape chemotherapy [2]. Importantly, this can also compromise bone remodelling, a key process whereby osteoblasts and osteoclasts continuously remould and reshape the bone matrix in units known as bone remodelling compartments to maintain structural integrity [3]. Indeed, impaired bone remodelling can lead to the manifestation of secondary bone conditions commonly seen in patients with leukaemia, including osteolytic lesions, osteoporosis, and increased fracture risk [4,5]. Although the niche microenvironments of acute myeloid leukaemia (AML) and chronic myeloid leukaemia (CML) are well established, others such as acute lymphoblastic leukaemia (ALL) and chronic lymphocytic leukaemia (CLL) have not been thoroughly investigated, in part due to the lack of reliable animal models.

This review provides a current understanding of the BMM and the mechanisms involved in the maintenance of the leukaemic niche. In addition, we provide perspective on current BMM-targeted therapies for the treatment of myeloid and lymphoid malignancies and discuss the strengths and limitations associated with these treatment approaches. Knowledge gained from these studies will lead to improved treatment strategies and clinical outcomes for patients with high-risk leukaemia.

## 2. The Bone Marrow Niche

Studies conducted using various mouse models have delineated BM niches as endosteal or central, depending on their anatomical location within the BM structure [6]. The endosteal niche consists of osteolineage cells (e.g., osteoprogenitors, osteoblasts, osteoclasts) and mesenchymal stem cells (MSCs), which are localised on the inner bone lining surface and supported by the endosteal vessel network [7,8]. Insight into the modulation of HSCs by osteoblasts has been investigated using mouse models [9]. The mechanisms of osteoblast–HSC interaction have been validated as direct cell–cell contact (e.g., Notch1/Jagged1) and paracrine interactions via secretory factors (e.g., osteopontin and angiopoietin-1) to promote HSC maintenance, retention, and quiescence [9,10,11]. In addition, osteoblasts promote the maintenance of early lymphoid progenitors by producing C-X-C motif chemokine 12 (CXCL12), a prominent chemoattractant of haematopoietic cells [12]. This finding endorses the concept of osteoblasts and the overall endosteal niche support for haematopoiesis. In contrast, the central niche comprises vasculature and perivascular stromal cells, such as leptin receptor^+^ MSCs and CXCL12-abundant reticular cells [6]. These perivascular stromal cells, together with endothelial cells, are actively involved in HSC maintenance through local secretion of cytokines [13,14]. Importantly, these cells also regulate B lymphopoiesis through interleukin 7 production [15].

The dynamic nature of the BMM is attributable to a myriad of cytokines, chemokines, and adhesion molecules, which support crosstalk between haematopoietic and BM niche cells, resulting in healthy production and replenishment of myeloid and lymphoid cells. However, the presence of malignant cells outcompetes normal HSCs, leading to “hijacking” of the BM niche. Indeed, gradual changes of healthy BMM into a leukaemic niche or “sanctuary” is an important hallmark of leukaemogenesis, whereby leukaemic cells establish their own dynamic ecosystem and promote dormancy, survival, proliferation, and resistance to chemotherapy (Figure 1). The structure of the normal HSC niche [6,7,8], as well as cell-intrinsic/extrinsic niche factors that regulate clonal haematopoiesis and leukaemia progression [16,17], have been extensively reviewed. Thus, this review will primarily focus on BMM-targeted therapies.

## 3. Leukaemia Niche Regulation

### 3.1. Niche-Driven Malignant Transformation

Significant progress has been made over recent decades in understanding how genetic mutations or functional alterations of niche cells can promote leukaemogenesis. Deletion of certain genes (e.g., *Rarg*, *Rb1*, and *Mib1*) in non-haematopoietic cells is a major risk factor for the induction of myeloproliferative neoplasia (MPN)-like disease, a pre-leukaemic disorder, in several mouse models [6]. Deletion of *Dicer1* in osteoprogenitors has been shown to result in myelodysplastic syndrome (MDS) with sporadic transformation to AML [18]. Apart from loss-of-function, constitutive activation of certain genes can also promote leukaemogenesis. For instance, osteoblast-associated activating mutation of β-catenin, present in 38% of patients with MDS or AML, can promote the development of leukaemia [19]. Likewise, activation of the parathyroid hormone receptor in osteoblasts attenuates BCR-ABL1-induced CML-like MPN and enhances *KMT2A-MLLT3* oncogene-induced AML, providing further evidence that osteoblasts are capable of influencing haematological malignant transformation [20]. Endothelial cells have also been implicated in the development of MPN-like disorder through miR-155 microRNA-induced nuclear factor κB (NF-κB) activation and pro-inflammatory cytokine production [21]. The oncogenic role of MSCs has recently been further highlighted, where activating *Ptpn11* mutations in MSCs and osteoprogenitors induce MPN through excessive production of chemokine ligand 3 (CCL3) [22]. This evidence points towards malignant transformation being dependent on the oncogenic tendencies of the surrounding cells, with such predisposition contributing to establishment of the BMM as a fertile ground for leukaemogenesis.

### 3.2. Leukaemic Remodelling of the Vasculature and Endosteal Niche

Leukaemic cells are also capable of remodelling the BMM into a cancer-supportive environment, known to facilitate tumour survival, resistance to therapy, and immune escape. AML cells co-expressing BCR-ABL1 and Nup98/HoxA9 fusion gene have been shown to disrupt bone homeostasis by inhibition of mature osteoblasts, likely via leukaemic cell-secreted CCL3 in vivo [23]. Similarly, *KMT2A-MLLT3* AML cells impair HSC niche function and induce commitment of MSCs to differentiate into osteoprogenitor cells but not mature osteoblasts, leading to reduced bone mineralisation in vivo [24]. Furthermore, a study using a BCR-ABL driven CML/MPN mouse model showed that leukaemic myeloid cells remodelled the endosteal BM niche by promoting aberrant expansion of osteoblastic lineage cells [25]. These cells exhibited compromised HSC-supportive activity and favoured abnormal BM myelofibrosis, which is often associated with poor prognosis in CML [25]. Recently, T-ALL cells have also been implicated in inducing osteoblast apoptosis and impairing haematopoiesis [26] Further study revealed that the mechanism of T-ALL-mediated osteoblast suppression is via aberrant Notch activation, which plays a role in negatively regulating CXCL12 on osteoblasts [27].

Leukaemic cells are also known to cluster around BM blood vessels, where AML engraftment directly contributes to an altered vascular architecture, increased endothelial cells, and increased vascular permeability [28]. This finding converges on a model whereby the vascular niche, similar to the endosteal niche, is being remodelled by leukaemic cells to favour leukaemogenesis. Several studies have associated increased vascularity in leukaemic BM with the production of angiogenic factors and inflammatory cytokines. AML cells secrete vascular endothelial growth factor A (VEGF-A), which promotes both leukaemic cell proliferation and tumour-supportive angiogenesis [29]. Increased plasma VEGF levels in patients with AML has been associated with poor clinical outcome [30]. Activation of endothelial cells by VEGF-A can also contribute to increased leukaemic cell protection by the vascular niche, allowing the cells to escape from therapy-induced cytotoxicity [31]. Furthermore, AML engraftment can lead to the formation of a hypoxic niche microenvironment, alter the molecular signature of endothelial cells, and promote the production of hypoxia-responsive reactive oxygen species and nitric oxide (NO), a mediator of vascular permeability [28]. Importantly, NO-induced permeability and vessel leakiness are associated with a dysregulated blood supply and poor delivery of therapeutic agents to malignant target cells [28].

Remodelling of the vasculature by leukaemic cells is not strictly confined to the vascular sinusoids located within the central marrow, but also the endosteal vessels. Indeed, endosteal remodelling by AML can lead to the loss of blood vessels, correlating with progressive depletion of stromal cells, osteoblasts, and HSCs [32]. This striking phenotype indicates area-specific changes with differential remodelling of vessels in central and endosteal BM regions. Further characterisation of the mechanisms revealed that endosteal AML cells produced pro-inflammatory signals such as tumour necrosis factor and anti-angiogenic cytokines, which play a major role in local suppression and destruction of endosteal vasculature [32]. In addition, emerging evidence also suggests that leukaemic cell-secreted inflammatory mediators are pivotal in regulating the expression of endothelial adhesion molecules (e.g., E-selectin) and initiate leukaemic–endothelial interaction, which in turn modulates leukaemogenesis in a positive feedback loop mechanism [33,34]. Collectively, the mechanisms associated with BM vascular remodelling by leukaemic cells represent an important approach for therapeutic targeting in haematological malignancies.

Our recent understanding of the BMM in driving leukaemia manifestations and therapy resistance has significantly advanced due to investigations using a range of transgenic, syngeneic, and patient-derived xenograft leukaemia engraftment models. While most studies have investigated the microenvironment of myeloid malignancies, emerging evidence points to specificity of microenvironment remodelling for different types of leukaemia. For instance, endosteal vessels were found to be decreased in AML-burdened mice but were unaffected by T-ALL [26,32]. Furthermore, the manifestation of bone loss in pre-B ALL is different from AML: BCR-ABL1^+^ pre-B ALL mice have been shown to develop progressive but severe bone loss during leukaemogenesis caused by impaired osteogenesis, coupled with enhanced osteoclastogenesis and bone resorption via excessive production of RANKL by leukaemic cells [35]. Further characterisation of the pre-B ALL BMM using single-cell RNA sequencing revealed significant impairment of the haematopoietic stem and progenitor cells (HSPCs) compartment, intercellular communication changes in monocytes and HSPCs, and alteration of molecular signatures associated with pro-B and pre-B cell responses [36]. The different leukaemia-specific mechanisms of remodelling remain significant areas of interest for further investigation.

### 3.3. Adipocytic Niche in Leukaemia

Adipocytes are one of the key cellular components of the BMM. They occupy the majority of the BM space and play an important role in regulating normal haematopoiesis. Accumulating evidence also suggests that BM adipose tissues can impact on the survival and proliferation of leukaemic cells in AML and T-ALL [37,38]. Furthermore, adipocytes are also known to contribute to leukaemic chemoresistance by metabolising and inactivating drugs in the BM [39]. Thus, the concept of targeting adipocytes in the leukaemic BMM remains an attractive proposition due to their influence as a protective niche for malignant cells and has been extensively reviewed [40,41].

## 4. Targeting the Leukaemia Microenvironment: Mechanisms of Treatment Resistance and Therapeutic Strategies

Recognising that the tumour microenvironment contributes to treatment failure or success has led to a recent paradigm shift in cancer therapy. In leukaemia, targeting components or niche-associated signalling pathways of the disease microenvironment could present a novel means of enhancing therapeutic effectiveness, particularly in children with high-risk leukaemia, as dose-limiting toxicities of conventional chemotherapeutic agents have prevented further improvements in survival. As some BM niche factors involved in the survival mechanism of leukaemic cells are also major regulators of normal HSC mobilisation/trafficking [42], therapeutically targeting these factors to modulate the mobilisation of leukaemic cells could improve clinical outcomes. The therapeutic agents described and their mechanisms of action within the leukaemia BMM have been summarised in Table 1 and illustrated in Figure 2. The clinical trial outcomes for each therapeutic agent have been summarised in Table 2.

### 4.1. CXCL12/CXCR4 Signalling Pathway

The C-X-C motif chemokine 12/C-X-C chemokine receptor type 4 (CXCL12/CXCR4) axis has been highlighted as a key driver of chemoresistance in leukaemia. In AML cells, chemotherapy has been reported to upregulate CXCR4 expression, thereby conferring resistance to therapy-induced apoptosis when co-cultured with stroma cells [105]. Likewise, stromal CXCL12 is required for homing, retention, and repopulation of pre-B ALL cells in the BM [106]. Interestingly, direct contact of T-ALL cells with CXCL12-producing endothelial cells has been implicated in the formation of a vascular niche for T-ALL [50].

Numerous preclinical studies have demonstrated that CXCR4 inhibitors can improve treatment strategies for leukaemia. Plerixafor (AMD3100) is a potent, U.S. Food and Drug Administration (FDA)-approved bicyclam CXCR4 antagonist for the treatment of multiple myeloma and lymphoma. In vivo studies have shown that plerixafor can mobilise and sensitise leukaemic cells to the chemotoxicity of treatment by disrupting CXCL12/CXCR4 interactions between leukaemic cells and the stromal niche [43,44]. The safety and tolerability of plerixafor has been demonstrated in phase 1 trials for patients with relapsed/refractory acute leukaemia and in patients with AML undergoing allogeneic haematopoietic stem cell transplantation (HSCT) [46,47,48]. Other derivatives of plerixafor, such as AMD3465 and AMD11070, have also been reported to elicit anti-leukaemic activity. Treatment with AMD3465 in vivo led to the reduction of leukaemic burden in pre-B ALL and T-ALL, while AMD3465 was shown to antagonise stroma-induced migration of AML and enhance the susceptibility of leukaemic cells to chemotherapy [45,49,50]. Likewise, AMD11070 has been shown to enhance the therapeutic efficacy of chemotherapy in mice transplanted with pre-B ALL cells [51]. Whilst plerixafor is the only reported bicyclam CXCR4 inhibitor to have undergone clinical assessment, in vivo data suggest further clinical investigation of others is warranted.

In recent years, synthetically designed peptides have emerged as another group of promising agents owing to their high-affinity for CXCR4 binding and ability to mobilise leukaemic cells from their protective niche. BL-8040 has been shown to induce leukaemic cell apoptosis in both AML-bearing mice and patients [52,53]. Identification of another potent CXCR4 peptide antagonist, LY2510924, led to a phase 1 study that demonstrated safety and yielded a response signal in patients with relapsed/refractory AML, supporting further clinical investigation [54,55]. Other emerging CXCR4 peptide antagonists capable of mobilising leukaemic cells include E5 and POL6326, with the latter currently under clinical evaluation (NCT01413568) [57,58]. Interestingly, low anticoagulant 2-O, 3-O desulphated heparin (CX-01), an analog of heparin with reduced anticoagulation, has been suggested as a potential therapeutic agent due to its CXCL12/CXCR4-inhibitory properties [63]. Encouraging complete remission rates were seen when CX-01 was administered to patients with AML in combination with standard induction therapy, with disruption of CXCL12/CXCR4-mediated protection of leukaemic stem cells (LSCs) in the BM niche playing a potential role in treatment response [63].

Another CXCL12/CXCR4-targeted therapeutic strategy involves the use of monoclonal antibodies. Ulocuplumab (BMS-936564/MDX-1338) is a first-in-class human IgG4 monoclonal anti-CXCR4 antibody that has therapeutic potential for several haematologic malignancies [59]. Preclinical studies have shown that ulocuplumab inhibits CXCL12-induced cell migration, induces apoptosis in leukaemic cells and exhibits anti-tumour effects when used as monotherapy in AML mouse xenograft models [59]. Moreover, clinical studies have revealed the ability of ulocuplumab to mobilise leukaemic cells into the circulation, rendering them more susceptible to chemotherapy-induced toxicity in patients with AML [60,61]. LY2624587, another humanised CXCR4 antagonistic antibody, was found to elicit dose-dependent apoptosis of CCRF-CEM T-ALL in vitro and tumour growth inhibition in vivo [62]. However, the clinical trial status of LY2624587 in leukaemia is currently unknown.

### 4.2. Notch Signalling Pathway

Notch signalling is another important mechanism that contributes to leukaemia progression. Abnormal Notch1 activating mutations are found in over 50% of T-ALL cases, and various studies have shown that stromal-induced maintenance and growth of human T-ALL cells, and BM engraftment of T-ALL, are initiated by Notch1 activation [107,108]. In addition, pre-B ALL samples collected from high-risk patients showed elevated expression of Notch3/4 and Jagged2, and that Notch 3 and 4 signalling pathways were responsible for a stromal-mediated anti-apoptotic effect on pre-B ALL cells [109,110]. Furthermore, the role of Notch signalling has also been recognised in the pathogenesis of AML. Bone marrow MSCs (BM-MSCs) isolated from patients with AML expressed higher levels of Notch1 and Jagged1 proteins compared to healthy donors and promoted the proliferation, survival, and chemoresistance of AML cells [111]. In the vascular niche, endothelial cells also interact with T-ALL cells in a Notch-mediated fashion, thus promoting tumour growth and survival [112].

The relevance of Notch signalling in the crosstalk between leukaemic cells and BM stromal/MSCs render this pathway as attractive therapeutic target. One such strategy is the use of monoclonal antibodies with targeted specificity against extracellular Notch receptors or ligands. Antibody-blocking Notch ligand DLL4 has been shown to impair tumour growth and enhance apoptosis in T-ALL, partly through the modulation of stromal cells [113]. While humanised anti-DLL4 antibodies such as demcizumab and enoticumab have undergone clinical trials for solid tumours, neither have been tested in haematological malignancies [114,115]. The gamma–secretase complex plays a pivotal role in the activation of Notch signalling. During receptor–ligand interaction, Notch receptors are cleaved by ADAM/TACE metalloproteinases, followed by another cleavage by the gamma–secretase complex to induce Notch activation [116]. Therefore, inhibiting this crucial step with gamma–secretase inhibitors (GSIs) is an appealing treatment approach for haematological malignancies. Treatment of pre-B ALL-bearing mice with GSI-XII resulted in enhanced chemo-sensitivity of ALL cells, leading to reduced BM leukaemic burden and prolonged survival [110]. Another GSI, MK-0752, was clinically evaluated in seven patients with T-ALL and one with AML [117]. While MK-0752 was poorly tolerated due to gastrointestinal toxicities caused by the “on-target” abrogation of Notch1/2 in the gut, such adverse effects could be overcome by combining the therapy with glucocorticoids and adopting an intermittent dosing regimen [118,119]. Indeed, minimal gastrointestinal toxicity of this treatment regimen was verified using PF-03084014, a GSI in clinical development [64]. PF-03084014, in combination with fludarabine, has been shown to exhibit synergistic anti-tumour effects against Notch1-mutated CLL cells in vitro, even in the presence of protective stromal cells [120]. Furthermore, PF-03084014 synergistically enhanced glucocorticoid-induced reduction of tumour burden in T-ALL-bearing mice [65]. A subsequent phase 1 trial has demonstrated promising anti-tumour activity in patients with T-ALL, thus supporting the potential use of PF-03084014 in the clinical setting [66]. Another promising GSI currently under clinical development is BMS-906024. In a phase 1 trial, 32% of patients with relapsed/refractory T-ALL showed at least 50% reduction in BM blasts, and treatment with BMS-906024 was generally well tolerated [68]. MRK-560 is a highly selective GSI that inhibits the presenilin-1 (PSEN1) subclass of gamma–secretase complexes, which has high expression in T-ALL. A recent study has demonstrated that MRK-560 treatment can impair leukaemogenesis and prolong survival in T-ALL bearing mice without inducing gastrointestinal toxicity, indicating a potential therapeutic advantage over other broad-spectrum GSIs [69].

### 4.3. Wnt/β-Catenin Signalling Pathway

Aberrant activation of Wnt/β-catenin signalling is a major inducer of survival and recurrence in many cancers, including leukaemia. A recent genomic analysis of relapsed childhood ALL has identified the Wnt/β-catenin pathway as one of the top dysregulated pathways [121]. Aberrant activation of Wnt/β-catenin signalling, either through methylation of Wnt antagonists and/or mutations in Wnt signalling regulatory proteins, is known to cause disease progression in AML [122]. While cell-intrinsic mechanisms of Wnt signalling are directly involved in malignant transformation, compelling evidence points to the BM niche as an extrinsic activator of Wnt signalling and leukaemogenesis. For instance, leukaemogenesis initiated by osteoblasts is modulated by Notch/Jagged1 interactions in HSCs, and Jagged1 expression in osteoblasts is stimulated by β-catenin activation through interaction with the transcription factor, FoxO1 [123]. BM stromal cells also contribute to the chemoprotective effect of leukaemic cells via the Wnt signalling pathway [70]. Intriguingly, Wnt signalling has also been implicated in the pro-survival regulation of AML cells by the BM vascular niche [124]. E-selectin, an endothelial-specific adhesion molecule and regulator of HSCs, has been reported to interact with AML cells and enhance leukaemic survival via Wnt activation [124]. In addition, the expression of BM E-selectin was found to be increased during AML development, which correlated with increased E-selectin-binding potential of AML blasts. This suggests that multiple niche-derived factors, including Wnt, could be involved in the complex interaction between leukaemic cells and niche resident cells to support leukaemia cell attachment and retention.

While there are currently no known FDA-approved Wnt-targeting agents, several investigational drugs have been assessed in preclinical and clinical studies. XAV939, a Wnt signalling antagonist, exhibited anti-leukaemic activity by inhibiting AML cell viability and attenuating MSC-induced chemoprotective effects of ALL cells in vitro [70,125]. In addition, XAV939 enhanced chemotherapy-induced apoptosis in ALL and improved overall survival of ALL-bearing mice [70]. PRI-724, a β-catenin inhibitor currently undergoing clinical evaluation (NCT01606579), has also demonstrated anti-leukaemic activity in AML and CML. In a blast crisis CML-bearing mouse model, combined treatment of PRI-724 and nilotinib led to impaired engraftment of CML cells, prolonged survival of tumour-bearing mice, and synergistically induced cell death in tyrosine kinase inhibitor-resistant leukaemia cells [71]. Similarly, PRI-724 has been shown to synergistically enhance the anti-tumour effect of FLT3 tyrosine kinase inhibitors in vitro and in vivo by abrogating the BMM-induced protection of FLT3-mutated AML cells [72]. BC2059, an anthraquinone oxime-analogue, has been shown to synergistically induce apoptosis in primary AML blasts and prolong survival of AML xenograft mice following administration with panobinostat [73]. CWP232291, a novel peptidomimetic that possess broad preclinical anti-cancer activities, has recently been tested in a phase 1 trial in patients with AML, with minimal/modest efficacy [74,126,127]. Future studies have been planned to explore the potential synergistic effects of CWP232291 with other chemotherapeutic agents [74]. SKLB-677, a recently developed dual inhibitor of FLT3 and Wnt/β-catenin signalling pathways, was found to suppress the viability of FLT3-driven AML cells and prolonged the survival of AML xenograft mice in a dose-dependent manner [75].

Collectively, preclinical and clinical studies to date have supported the potential for Wnt/β-catenin inhibitors to treat leukaemia, largely due to their cytotoxicity on leukaemia cells and their ability to modulate the leukaemic microenvironment. However, the challenges of targeting the Wnt pathway have been highlighted by a study that found variable response to Wnt inhibition in primary blasts from different patients with normal karyotype AML, indicating that certain subtypes of AML may be more susceptible to anti-Wnt therapy, necessitating further research in this area [128].

### 4.4. Cell–Cell Adhesion

Direct cell–cell contact via adhesion molecules is an indispensable strategy not only for physiological BM homeostasis, but also for the survival of leukaemic cells. For instance, the interaction between lymphocyte function-associated antigen 1 expressed on T-ALL cells and its ligand, intercellular adhesion molecule 1 expressed on BM-MSCs, promotes leukaemic cell survival, highlighting its critical role in the T-ALL niche [129]. Another important adhesion molecule, very late antigen-4 (VLA-4) (α4β1 integrin), is known to be highly expressed on leukaemic blasts from children with relapsed pre-B ALL and is associated with poor prognosis and overall survival [130]. VLA-4 plays an integral role in mediating leukaemia cell adhesion and chemoresistance by interacting with vascular cell adhesion molecule 1 (VCAM-1) expressed on BM MSCs [131]. Importantly, VLA-4/VCAM-1 is also involved in mediating the adhesion of AML to endothelial cells, suggesting a role within the BM vascular niche [31].

Recognition of the VLA-4/VCAM-1 interaction within the leukaemia–BMM niche has led to development of numerous pharmaceuticals targeting this specific pathway. Natalizumab is a clinically available, humanised monoclonal anti-VLA-4 with anti-tumour potential in AML and pre-B ALL [78,79]. Mechanistically, natalizumab promotes mobilisation and chemosensitivity of leukaemic cells by inhibiting VLA-4-mediated stromal adhesion [78,79]. However, its widespread therapeutic use has been limited due to the risk of multifocal leukoencephalopathy [132]. Small molecules, such as TBC3486, have shown therapeutic potential by competing with VCAM-1 for integrin α4 binding, thereby sensitising pre-B ALL cells to chemotherapy [80]. FNIII14 peptide, another potent inhibitor of VLA-4, successfully eradicated minimal residual disease in the BM of mice with AML when combined with cytarabine, achieving a remarkable 100% survival rate [81].

CD44 is recognised as another key cell surface adhesion molecule that promotes the leukaemia–BMM niche interaction and BM engraftment of leukaemic cells [108]. Using a mouse model xenografted with haematopoietic progenitors expressing oncogenic Notch1 to drive T-ALL development, treatment with anti-CD44 monoclonal antibody was shown to antagonise leukaemic cell activity and prolong survival by impairing the interaction between leukaemic cells and stromal niches [108]. In addition, several other studies have supported the therapeutic potential of CD44-targeted antibodies for the treatment of AML [76,133]. RG7356 (or ARH460-16-2), a humanised anti-CD44 monoclonal antibody, has been investigated in a phase 1 dose-escalation study for patients with relapsed/refractory AML and was found to be safe and well tolerated [77].

Another emerging adhesion-mediated therapeutic target is E-selectin, an important contributor to chemoresistance and disease relapse. Uproleselan (GMI-1271) is an endothelial niche modulator and a small molecule mimetic that targets E-selectin [34]. A recent study has shown that inhibition of the E-selectin-mediated pro-survival signalling pathway with uproleselan suppressed AML blast regeneration and enhanced survival of leukaemia-bearing mice harbouring the human *KMT2A-MLLT3* oncogene via a synergistic effect with chemotherapy [34]. Encouragingly, an early phase study revealed that administration of uproleselan with chemotherapy led to high remission rates and promising survival outcomes in patients with relapsed/refractory AML [82]. Phase 2/3 clinical trials are currently underway to further evaluate the efficacy of uproselesan in AML (NCT03701308, NCT03616470). GMI-1359, a dual inhibitor of CXCR4/E-selectin, has been shown to efficiently mobilise leukaemic cells into the circulation and strongly extend survival of mice xenografted with FLT3-internal tandem duplication (FLT3-ITD) mutant AML cells either alone or in combination with the FLT3-ITD inhibitor sorafenib [83]. Strikingly, administration of GMI-1359 and sorafenib was also found to increase regeneration of normal BM haematopoietic cell populations [83]. As haematopoiesis is often perturbed during leukaemia development, a dual therapeutic strategy of eliminating leukaemic cells and restoring haematopoiesis constitutes an attractive strategy to improve clinical outcomes.

### 4.5. Bone Remodelling Signalling Pathways

Osteolytic lesions or pathological fractures are common features of dysregulated bone remodelling in certain cancers such as multiple myeloma and metastatic breast cancer, where the interactions between cancer cells and the BMM impair the homeostatic balance between bone formation and resorption, leading to bone loss and destruction. Despite manifestations of bone diseases in many patients with leukaemia, therapeutic clinical evaluation of bone anabolic agents is lacking. One promising candidate is zoledronic acid, an FDA-approved drug for the treatment of osteoporosis. Mechanistically, zoledronic acid directly inhibits farnesyl diphosphonate synthase in osteoclasts, thereby inhibiting resorption and survival [134]. Administration of zoledronic acid led to an improved pain profile in children with ALL who developed treatment-related osteonecrosis [135]. Furthermore, attenuating osteoclast-induced bone loss with zoledronic acid has been shown to reduce the leukaemia burden in the BM and extend survival in a syngeneic BCR-ABL1^+^ pre-B ALL mouse model [35]. These findings provide support for clinical investigation of zoledronic acid to restore the BMM in patients with leukaemia.

Studies investigating the use of bone anabolic agents as treatment applications for malignancies other than leukaemia have yielded positive outcomes, prompting further interest in their potential therapeutic efficacy for leukaemia. Interestingly, some of these agents have only been investigated in preclinical and clinical studies for their cytotoxicity against leukaemia. Everolimus, a mammalian target of the rapamycin (mTOR) inhibitor, is an inhibitor of osteoclast resorption and has been shown to prevent bone loss in ovariectomised mice [136]. The clinical efficacy of everolimus administration in combination with multi-agent chemotherapy has been evaluated in adult and paediatric relapsed/refractory ALL. Everolimus was shown to be safe and effective in T-ALL (50% response) when combined with chemotherapy [84]. Likewise, the combination of everolimus with four-drug reinduction therapy yielded favourable second complete remission rates and low levels of minimal residual disease following reinduction [85]. However, these studies did not assess bone parameters that could provide important validation regarding the effects of everolimus on leukaemia-induced bone loss. Another promising agent, cabozantinib, exhibits potent activity against multiple receptor tyrosine kinases that modulate tumour survival and metastasis [137]. Notably, cabozantinib is also capable of remodelling the BMM through inhibition of osteoclast differentiation and resorption, as well as modulating the RANKL/osteoprotegerin ratio in osteoblasts [138]. In a recent preclinical evaluation, cabozantinib was found to selectively induce cytotoxic effects in AML cells with FLT3-ITD in vivo [86]. Importantly, a dose-escalation study demonstrated that cabozantinib was well tolerated in patients with AML, thus establishing a precedent for larger efficacy trials in the future [87].

The ubiquitin–proteasome pathway also presents as an attractive target due to its role in the degradation of regulatory proteins, including cell cycle proteins, tumour suppressor proteins, and transcription factors associated with bone metabolism [139]. Bortezomib is an FDA-approved proteasome inhibitor that can normalise bone remodelling in multiple myeloma by exerting inhibitory effects on osteoclast activity and stimulatory effects on osteoblast differentiation/growth [140]. A preclinical study found that bortezomib exerted an anti-tumour effect on human T-ALL by inhibiting Notch1 target genes [88]. Similarly, bortezomib has been shown to potentiate volasertib-induced mitotic arrest in AML cells by inhibiting slow degradation of cyclin B and prolonged survival of volasertib-treated AML-bearing mice [89]. Several clinical studies have evaluated bortezomib in combination with four-drug induction therapy for relapsed ALL. Although these studies have shown an improvement in response rate, a survival advantage is yet to be elucidated [90]. The addition of bortezomib to standard chemotherapy has also failed to improve survival in children with AML [91]. A number of second-generation proteasome inhibitors have subsequently been developed. These include carfilzomib and ixazomib, both of which are capable of exerting bone anabolic activity via dual effects of promoting osteoblastic bone formation and suppressing osteoclastic resorption [141,142,143]. Carfilzomib has also been shown to induce leukaemia cell apoptosis and achieved modest anti-leukaemic activity in a phase 1 study in patients with AML [92,94]. Several early phase clinical trials examining the anti-leukaemic effect of carfilzomib are currently recruiting (NCT02303821 and NCT02512926) [95]. A recent pre-clinical study by our group demonstrated that cafilzomib treatment did not confer a survival advantage when combined with multi-agent induction chemotherapy of vincristine, dexamethasone, and L-asparaginase, although the impact of treatment on bone was not examined [93]. Likewise, an encouraging response rate was seen in a phase 1 clinical trial that evaluated the efficacy of ixazomib administered in combination with mitoxantrone, etoposide, and cytarabine [96]. However, due to limited in vivo assessment, it remains unclear whether proteasome inhibitors could reverse defective bone remodelling in the leukaemia BMM.

### 4.6. Hypoxia-Related Signalling Pathways

In the BMM, low oxygen distribution provides a hypoxic niche where leukaemic cells preferentially reside, contributing to chemoresistance [144]. Furthermore, activation of hypoxia-inducible factor 1α (HIF-1α) in leukaemic cells, either via hypoxia-induced stabilisation or in response to oncogenic stimuli, can facilitate angiogenesis and production of pro-survival factors that favour chemoresistance [145,146]. However, the option of therapeutically targeting HIF-1α remains controversial in light of recent findings that showed that the loss of HIF-1α could accelerate leukaemogenesis [147,148]. In this context, an emerging group of therapeutic candidates known as hypoxia activated prodrugs, which are inert in normoxia and active only in hypoxia, are being evaluated for treatment of leukaemia.

PR-104 is a phosphate ester prodrug that can be converted into alcohol PR-104A in vivo and act as a hypoxia-activated prodrug [149]. Mechanistically, PR-104A is selectively reduced to reactive cytotoxic nitrogen mustards (hydroxylamine PR-104H and amine PR-104M) upon encountering a hypoxic environment and induces cell cytotoxicity via DNA cross-linking [149]. PR-104 has been shown to induce hypoxia-selective growth inhibition of pre-B ALL cells in vitro, and the administration of PR-104 in AML and ALL mouse xenografts reduced leukaemic burden and prolonged survival [97]. Another reported mechanism of PR-104 involves hypoxia-independent activation by the enzyme aldo-keto reductase 1C3 (AKR1C3) [150]. Primary T-ALL cells have been found to be more sensitive to the cytotoxicity of PR-104 compared to pre-B ALL cells due to higher expression of AKR1C3 [98]. In a phase 1/2 clinical trial of relapsed/refractory ALL and AML, PR-104 was found to exert measurable clinical activity [99]. However, significant toxicity was also observed in some participants, likely due to the higher dose administered compared to the dose used in solid tumour trials. Further clinical evaluation of PR-104 should consider lower doses and tolerability in combination with chemotherapeutic agents. Another extensively examined hypoxia-activated prodrug is evofosfamide (TH-302), a 2-nitroimidazole prodrug of the cytotoxin bromo-isophosphoramide mustard, which has shown efficacy in a range of different tumour models [151]. The anti-tumour effect of evofosfamide has been demonstrated in primary leukaemia samples in vitro, as well as in vivo, where evofosfamide treatment of AML xenografts reduced disease burden and prolonged survival without overt haematological toxicity [100,101]. A phase 1 study revealed that evofosfamide therapy significantly suppressed hypoxia markers in the leukaemic BM; however, there was limited activity in heavily pre-treated patients with leukaemia, which may hamper further clinical investigation [102].

### 4.7. The Vasculature

Another treatment approach that has been extensively investigated is the use of anti-angiogenic agents that target neovascularisation within the leukaemic niche. However, clinical outcomes have largely been disappointing due to significant adverse effects and limited response rates in patients with leukaemia. For instance, VEGF receptor inhibitors such as vatalanib and sorafenib have not been reported to confer additional survival benefit for myeloid neoplasms in adult and elderly patients, even when used in conjunction with chemotherapy [152,153]. However, sorafenib’s potential benefit has been noted in FLT3/ITD^+^ AML patients, which may be related to its direct anti-leukaemic effect [154]. An alternative approach is the use of vascular disrupting agents, which can induce breakdown of the vascular architecture, leading to disruption of the vascular supply to tumour cells [155]. Combretastatin-A1-diphosphate (OXi4503/CA1P), a microtubule-destabilising agent, has been shown to induce vascular disruption, delay tumour growth, and increase survival in HL60 leukaemia-bearing mice [103]. In addition, a recently completed clinical study demonstrated that CA1P administered with cytarabine showed early clinical activity and was well tolerated by patients with relapsed/refractory AML [104]. CA1P has been granted Rare Paediatric Disease designation by the FDA for children with AML, paving the way for further future clinical investigation.

The therapeutic approach of targeting angiogenesis within the leukaemia microenvironment is further complicated by differential remodelling of the vasculature, depending on the niche location [32]. In AML, leukaemic infiltration results in a loss of endosteal vessels. A vessel-poor microenvironment or abnormal vessels incapable of supplying oxygen or nutrients could hinder the efficiency of drug delivery and thus promote chemoresistance. Therefore, a niche-specific strategy is needed to overcome the non-specificity of targeting the BM vasculature. In this context, agents that can rescue endosteal vessels prior to chemotherapy could be highly beneficial for patients. For instance, deferoxamine has been shown to promote HSC homing and endosteal vessel formation in AML-bearing mice, which could potentially confer additional treatment benefits [32]. Thus, clinical studies are needed to investigate the potential of incorporating an endosteal antagonist into conventional chemotherapeutic regimens to improve survival outcomes.

### 4.8. Emerging Therapies: Immunotherapies and Mitochondrial-Targeted Therapies

In recent years, the immune cell microenvironment of haematologic malignancies has been increasingly recognised as an important target for immunotherapies. In particular, tumour-associated macrophages (TAMs) are capable of being reprogrammed by tumour cells into an immunosuppressive and pro-tumourigenic phenotype, thus contributing to subdued immune responses and promoting disease progression [156]. Immune-based therapies that target macrophages in haematologic malignancies have been extensively reviewed [157]. In particular, CD47 (a surface protein that is expressed by leukaemic cells) has emerged as a key target for immunotherapy due to its role in innate immune evasion by inhibiting macrophage phagocytosis [158], with a recent review reporting current clinical trials involving the use of CD47 antagonists in haematologic malignancies [156]. The feasibility and effectiveness of other emerging immune-based therapies, such as genetically-engineered T-cells expressing a chimeric antigen receptor (CAR T-cell therapy), has been demonstrated in clinical trials, inspiring new and innovative approaches to management of leukaemia [159].

Finally, intercellular mitochondrial transfer through tunneling nanotubes and extracellular vesicles is regarded as a new and exciting crosstalk mechanism between BMM cells and leukaemic cells. Functional exchange of mitochondria between donor BM-MSCs and recipient leukaemic cells contribute to enhanced leukaemic cell metabolism, increased proliferative capacity, drug resistance, and disease relapse [160,161]. Therefore, targeting molecular components involved in mitochondrial transfer, such as CD38, could present a novel avenue for elimination of minimal residual disease in treatment for leukaemia [161]. Other promising mitochondrial-targeted approaches have been extensively reviewed elsewhere [162].

## 5. Open Questions and Future Perspectives

In recent years, extensive use of preclinical models has provided valuable insights into the pathophysiology of leukaemia. Through research, it is now clear that leukaemia is no longer viewed solely as a spontaneous cancer induced by genetic defects, but rather a disease highly dependent on both intrinsic and extrinsic mechanisms exerted by BM niche cells. Conversely, the BMM can also be remodelled into an oncogenic sanctuary by leukaemic cells to provide significant survival advantage and therapy resistance. Crosstalk between the leukaemic cells and BMM are highly dependent on cell–cell interactions and pathways, including CXCL12/CXCR4, VLA-4/VCAM-1, Notch, and Wnt, amongst others. As these mechanisms are often associated with haematopoietic niche regulation, leukaemia-induced changes in the BMM will compromise healthy haematopoiesis and blood cell turnover, ultimately leading to clinical manifestations of the disease.

Due to the indispensable role of the BMM in leukaemogenesis, targeting the sheltered leukaemia niche presents a feasible strategy to improve treatment outcomes. However, this approach has its own share of challenges, and effective therapeutic targeting has often remained elusive. In many cases, significant toxicities and limited efficacies exerted by the therapeutic agents in clinical trials urgently need to be addressed. For instance, a recent phase 3 clinical trial demonstrated that bortezomib increased toxicity but failed to improve overall survival in a large cohort of children with AML [91]. Similarly, administering PR-104 in patients with relapsed/refractory AML and ALL resulted in significant adverse events, such as myelosuppression, febrile neutropenia, infection, and enterocolitis [99]. Intriguingly, preclinical survival data supported the potential use of bortezomib for treatment of AML [89], and a similar survival advantage of PR-104 was also demonstrated in preclinical models of ALL [97,98]. It is important to note that due to practical limitations on the size of preclinical studies, treatment-related adverse events that are rare in patients may not be easily identified in animals despite similar or even higher dose levels/duration of drug administration [163]. In addition, early phase clinical trials often enrol heavily pre-treated patients with relapsed/refractory disease, as opposed to the young treatment-naive animals used in preclinical settings. Thus, increasing preclinical sample sizes and/or developing new patient-derived xenograft models that recapitulate patients with relapsed/refractory leukaemia could improve predictions of adverse events in humans.

While phase 1 clinical studies are primarily designed to explore safety and tolerability and are not powered for efficacy, outcomes from these studies often provide an indication as to whether certain leukaemia types respond better than others. For instance, it was found that patients with T-ALL appeared to be better responders compared to those with pre-B ALL (50% vs. 39%) when treated with everolimus in combination with HyperCVAD chemotherapy [84]. Of note, the patients with T-ALL in this study were also heavily pre-treated with a median of four prior therapies compared to those with pre-B ALL who had received a median of one prior therapy. While these findings ought to be interpreted with caution as the sample size was small, it was found that everolimus could increase the levels of PTEN, a major negative regulator of PI3K/Akt/mTOR signalling, in patients with T-ALL who achieved a response [84]. In contrast, patients with B-ALL had elevated PTEN levels but did not respond. This interesting finding indicates that the mTOR/PTEN pathway may play a more central role in T-ALL leukaemogenesis, and therefore further investigation into targeting this pathway for T-ALL is warranted. Others also point to certain genetic features and molecular markers as reliable predictors of therapeutic responses. For example, correlative studies have found that patients with increased levels of *IFI30* and *RORα* were associated with improved response to ixazomib [96]. Furthermore, E-selectin ligand expression in LSCs correlated with improved response and survival following uproleselan treatment in relapsed/refractory AML [82]. This suggests that gene expression/transcriptome profiling may be useful for predicting which patients are likely to benefit from or exhibit resistance to treatments. As such, future clinical trials that investigate novel agents targeting the BMM should implement such approaches to help identify biomarkers predictive of response.

Recently, cell mobilising agents have been intensely investigated due to their ability to enhance chemotherapy effectiveness by sensitising the leukaemic cells from their protective niche. In particular, agents that target the CXCL12/CXCR4 and cell–cell contact pathways are at the forefront of clinical trials. Remarkably, agents such as plerixafor can also enhance the haematologic recovery post-myeloablative allogeneic HSCT [164]. Other agents such as GMI-1359 have been shown to improve haematopoiesis during treatment [83]. Indeed, haematopoietic restoration is a vital clinical outcome during treatment and for recovery of marrow function post-treatment, and clinical use of drugs that can simultaneously potentiate anti-leukaemic activity and haematopoiesis is lacking. Therefore, such promising drug candidates should be prioritised for evaluation in clinical trials in combination with conventional therapy.

Another treatment strategy that remains significantly under-investigated is the use of bone anabolic agents as a therapeutic approach to reverse leukaemia-induced osteoporosis in patients. While some agents tested in clinical trials for leukaemia are known to be modulators of osteoblasts and osteoclasts, none of the trials evaluated the bone-normalising potential of these agents. A proof of concept study demonstrated that inhibiting osteoclast resorption in pre-B ALL mice with zoledronic acid could achieve a dual clinical outcome: normalising bone loss and prolonging survival [35]. In this study, zoledronic acid did not directly affect survival of the pre-B ALL cells, and whether it is the osteoclasts that directly contribute to survival remain to be explored [35]. As zoledronic acid’s safety profile in children has been demonstrated, future clinical trials could evaluate its efficacy as an adjuvant therapy in conjunction with conventional chemotherapy in childhood ALL. Other RANKL antagonists, such as the monoclonal antibody denosumab, have also displayed a good safety profile and may present an alternative therapeutic option [165]. Interestingly, a leukaemia-induced reduction in osteoblast numbers contributed to bone loss in pre-B ALL-bearing mice [35]. The mechanisms by which the leukaemic cells impair osteoblasts, either through suppression of osteoblastogenesis and/or induction of osteoblast apoptosis, are hitherto unclear and remain to be verified. It is plausible to postulate that increasing osteoblastic bone formation through therapeutic means may hinder leukaemogenesis and restore healthy haematopoiesis.

Advances in modern treatments have led to significant improvements in the prognosis of patients with leukaemia. Nevertheless, high-risk subgroups, including hypodiploid pre-B ALL, infant *KMT2A*-rearranged ALL, relapsed T-ALL, and AML, continue to have inferior outcomes. While older children, adolescents, and young adults are better able to tolerate intensified therapy, applying the same approach to infants and older adults is challenging. Over the years, infant-specific collaborative trials have made tremendous advances in optimising treatment protocols; however, despite this progress, infants with *KMT2A*-rearranged ALL continue to demonstrate inferior outcomes [166]. For older adults with AML, survival outcomes remain poor. Thus, we are approaching a therapeutic plateau and novel agents are desperately required to improve clinical outcome. Hence, the potential synergistic benefits of BMM-targeting agents with existing therapy should be explored more broadly, particularly in high-risk leukaemias that are difficult to treat. In this regard, well-characterised patient-derived xenograft models of high-risk leukaemia should be used for the evaluation of novel therapies due to their ability to mimic human disease and consideration as the “gold standard” models for preclinical testing. Future clinical trials should focus on integrating innovative therapies, identifying optimal dosing regimens and predictive biomarkers to maximise trial success.

## 6. Summary

The development of novel niche-targeted therapeutic agents currently has a cellular and pathway-specific focus, incorporating crosstalk disruption and/or direct cellular cytotoxicity as the main anti-leukaemic mechanisms. However, the strategy of blocking signalling pathways often carries the risk of counterproductive and unpredictable adverse events. Indeed, these limiting factors often act as roadblocks for clinical development of an investigational agent on its path towards FDA-approval. Therefore, it is important to examine the efficacy of these agents with conventionally used chemotherapeutic agents in clinical trials, deciphering the optimal doses and drug combinations to eradicate mobilised leukaemic cells without increasing toxicity. Finally, novel crosstalk mechanisms are constantly being discovered, such as leukaemic cell-derived exosomes and tunnelling nanotubes, thus opening up new and exciting avenues for therapeutic targeting. Future studies, combining aspects of both basic and clinical translational research, will facilitate better understanding of the pathological mechanisms within the leukaemia microenvironment, ultimately leading to the development of improved therapeutic approaches and outcomes for patients with leukaemia.

## Figures and Tables

**Figure 1 ijms-22-06888-f001:**
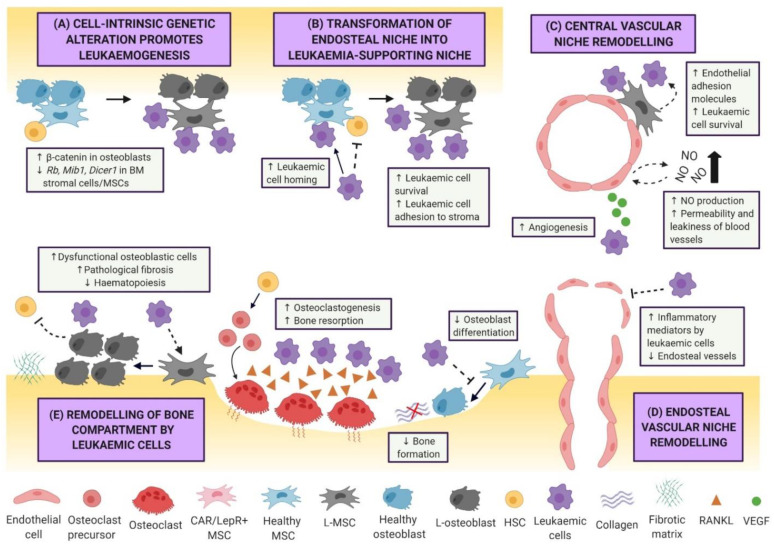
Leukaemogenesis and remodelling of the bone marrow niche by leukaemic cells. (**A**) Oncogenic transformation of haematopoietic cells into leukaemic cells can be initiated by cell-intrinsic genetic alterations in surrounding bone marrow niche cells. (**B**) During leukaemogenesis, malignant cells can hijack the mechanisms of homing used by healthy haematopoietic stem cells and directly compete with haematopoietic stem cells for endosteal adhesion, resulting in transformation of the healthy niche microenvironment into a tumour-supportive niche. Furthermore, leukaemic cells are capable of remodelling the (**C**) central vascular and (**D**) endosteal vascular niche into leukaemia-supportive microenvironments that favour leukaemia cell survival and expansion, with hostility towards physiological haematopoiesis. (**E**) Leukaemic cells can promote remodelling of bone compartments by directly influencing the differentiation and/or function of osteoblasts and osteoclasts, leading to severely impaired bone homeostasis.

**Figure 2 ijms-22-06888-f002:**
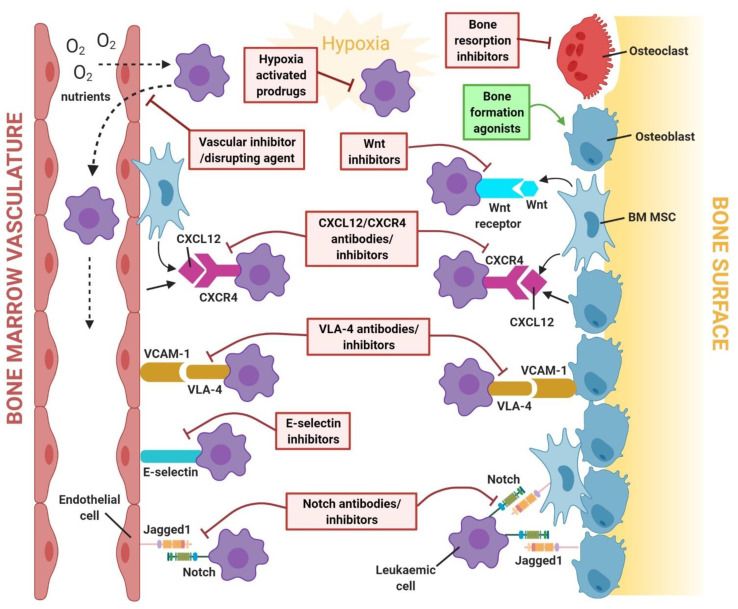
Regulatory and survival pathways targeted by therapeutic agents in the leukaemia bone marrow niche. Therapeutic agents can block interactions between leukaemic cells and niche cells by interfering with cell adhesion (e.g., VLA-4/VCAM-1, E-selectin, Notch/Jagged1) and paracrine regulation (e.g., CXCL12/CXCR4, Wnt/β-catenin). Furthermore, agents that are capable of normalising bone homeostasis (by targeting osteoblast and/or osteoclast activities), as well as targeting tumour-supportive vasculature and the hypoxic environment in the leukaemia niche, show therapeutic potential.

**Table 1 ijms-22-06888-t001:** Summary of therapeutic agents that target the leukaemic niche in the bone marrow microenvironment.

Target Pathways	Target Strategy	Agents	Mechanism of Action in the BMM	Leukaemia Type Investigated	Animal Studies	Clinical Studies
CXCL12/ CXCR4 signalling pathway	CXCR4 Inhibitor (bicyclams)	Plerixafor (AMD3100)	Induces mobilisation and chemosensitivity of leukaemic cells by targeting CXCL12/CXCR4 interactions.	AML, pre-B ALL	[43,44,45]	[46,47,48] *FDA-A*
		AMD3465	Induces mobilisation and chemosensitivity of leukaemic cells by targeting CXCL12/CXCR4 interactions.	AML, Pre-B ALL, T-ALL	[45,49,50]	N/R
		AMD11070	Induces mobilisation and chemosensitivity of leukaemic cells by targeting CXCL12/CXCR4 interactions.	Pre-B ALL	[51]	N/R
	CXCR4 inhibitor (synthetic peptides)	BL-8040 (BKT140)	Induces mobilisation and chemosensitivity of leukaemic cells by targeting CXCL12/CXCR4 interactions. Induces apoptosis in leukaemic cells.	AML	[52]	[53]
		LY2510924	Induces mobilisation and chemosensitivity of leukaemic cells by targeting CXCL12/CXCR4 interactions. Inhibits proliferation of leukaemic cells.	AML	[54]	[55]
		E5 Peptide	Induces mobilisation and chemosensitivity of leukaemic cells by targeting CXCL12/CXCR4 interactions. Induces apoptosis in leukaemic cells.	AML	[56,57]	N/R
		POL6326	Induces mobilisation and chemosensitivity of leukaemic cells by targeting CXCL12/CXCR4 interactions.	AML	[58]	NCT01413568Phase I/II #
	Anti-CXCR4 monoclonal antibody	Ulocuplumab	Induces apoptosis and blocks CXCL12-induced leukaemic cell migration. Induces mobilisation of leukaemic cells.	AML	[59]	[60,61]
		LY2624587	Induces apoptosis and blocks CXCL12-induced leukaemic cell migration.	T-ALL	[62]	N/R
	CXCL12 inhibitor	CX-01	Inhibits binding of CXCL12 to immobilised heparin and enhances treatment efficacy.	AML	N/R	[63]
Notch signalling pathway	Gamma-secretase inhibitor	PF-03084014	Inhibits stromal-mediated chemoresistance and potentiates sensitivity of leukaemic cells to chemotherapy. Inhibits proliferation and induces apoptosis in leukaemic cells.	T-ALL	[64,65]	[66]
		BMS-906024	Inhibits growth and survival of leukaemic cells by targeting Notch signalling.	T-ALL	[67]	[68]
		MRK-560	Promote leukaemic cell cycle arrest by targeting PSEN1, a subclass of gamma-secretase complexes highly involved in activation of mutant Notch1.	T-ALL	[69]	N/R
Wnt/β-catenin signalling pathway	Wnt/β-catenin inhibitor	XAV939	Attenuates BMM-induced protection of leukaemic cells by inhibiting Wnt signalling. Inhibits proliferation of leukaemic cells.	Pre-B ALL	[70]	N/R
		PRI-724	Attenuates BMM-induced protection of leukaemic cells by inhibiting Wnt signalling. Induces apoptosis in leukaemic cells.	AML, CML	[71,72]	NCT01606579Phase I/II #
		BC2059	Induces apoptosis of leukaemic cells by synergistically enhancing the effect of drug treatment.	AML	[73]	N/R
		CWP232291	Promotes endoplasmic reticulum stress activation, leading to degradation of β-catenin and apoptosis induction in leukaemic cells.	AML	N/R	[74]
	Wnt/β-catenin/FLT3 inhibitor	SKLB-677	Induces apoptosis in leukaemic cells.	AML	[75]	N/R
Adhesion molecules signalling pathway	Anti-CD44 monoclonal antibody	RG7356/ ARH460-16-2	Blocks leukaemia–stroma interaction by targeting CD44.	AML	[76]	[77]
	Anti-α4β1/ VLA-4 monoclonal antibody	Natalizumab	Blocks leukaemia–stroma interaction by targeting VLA-4/VCAM-1, sensitising leukaemic cells to chemotherapy.	AML, Pre-B ALL	[78,79]	N/R *FDA-A*
	α4 inhibitor	TBC3486	Blocks leukaemia–stroma interaction by targeting integrin α4, sensitising leukaemic cells to chemotherapy.	Pre-B ALL	[80]	N/R
	VLA-4 peptide antagonist	FNIII14	Blocks cell adhesion by targeting VLA-4 to fibronectin interaction, sensitising leukaemic cells to chemotherapy.	AML	[81]	N/R
	E-selectin inhibitor	Uproleselan (GMI-1271)	Attenuates cell surface adhesion, regeneration and survival of leukaemic cells by antagonising E-selectin. Sensitises leukaemic cells to chemotherapy.	AML	[34]	[82] NCT03701308 NCT03616470 Phase II/III #
	Dual CXCR4/ E-selectin inhibitor	GMI-1359	Promotes leukaemic cell mobilisation and restores normal haematopoiesis.	AML	[83]	N/R
Bone remodelling signalling pathway	Bone resorption inhibitor	Zoledronic acid	Inhibits osteoclast resorption.	Pre-B ALL	[35]	N/R *FDA-A*
	mTOR inhibitor	Everolimus	Inhibits osteoclast resorption. *	ALL	N/R	[84,85] *FDA-A*
	Receptor tyrosine kinase inhibitor	Cabozantinib	Inhibits osteoclast differentiation and resorption, modulates RANKL/osteoprotegerin in osteoblasts. * Induces apoptosis in leukaemic cells.	AML	[86]	[87] *FDA-A*
	Proteasome inhibitor	Bortezomib	Promotes osteoblast differentiation and suppress osteoclast activity. * Induces apoptosis in leukaemic cells.	AML, Pre-B ALL, T-ALL	[88,89]	[90,91] *FDA-A*
		Carfilzomib	Promotes osteoblast differentiation and suppress osteoclast activity. * Induces apoptosis in leukaemic cells.	AML, ALL	[92,93]	[94,95] NCT02303821 NCT02512926 Phase I *FDA-A*
		Ixazomib	Promotes osteoblast differentiation and suppress osteoclast activity. *	AML	N/R	[96] *FDA-A*
Hypoxia-related signalling pathway	Hypoxia activated prodrug	PR-104	Induces cytotoxicity in hypoxic leukaemic cells.	AML, Pre-B ALL, T-ALL	[97,98]	[99]
		Evofosfamide (TH-302)	Induces cytotoxicity in hypoxic leukaemic cells.	AML, ALL	[100,101]	[102]
Vasculature-associated pathway	Vascular disrupting agent	CA1P (OXi4503)	Induces breakdown of vascular architecture.	AML	[103]	[104]

N/R indicates not reported and status remains unclear. # ongoing clinical studies as described in the relevant reference or clinical trial number. * therapeutic mechanisms on bone remodelling confirmed using non-leukaemic animal models but remain to be confirmed in leukaemia animal models. *FDA-A* refers to agents clinically approved by the FDA for other disease treatments. ALL, acute lymphoblastic leukaemia; AML, acute myeloid leukaemia; BMM, bone marrow microenvironment.

**Table 2 ijms-22-06888-t002:** Summary of clinical trial outcomes for therapeutic agents that target the leukaemic niche in the bone marrow microenvironment.

Agent	Studies/ Clinical Trial ID (Phases)	Treatment Design	Age Range (Median)	Disease Type (No. Patients Enrolled)	Ref	Outcomes
Plerixafor	NCT01319864 (Phase 1)	Dose escalation at four different dose levels (6, 9, 12, and 15 mg/m^2^/dose) as part of cytarabine and etoposide therapy.	AML 3–17 (13) ALL 12–21 (14) AML/ MDS 20 (20)	R/R AML (13) R/R ALL (5) R/R AML/MDS (1)	[46]	Plerixafor mobilised leukaemic blasts into the peripheral blood. Combining plerixafor with cytarabine and etoposide was well tolerated. Most common agent-related non-haematologic AEs (grade ≥ 3) were febrile neutropenia and hypokalaemia. CR/CRi rate of 25% in patients with AML. Study not powered to define efficacy.
Plerixafor	NCT01141543 (Phase 1)	Dose escalation (240 μg/kg) as part of myeloablative conditioning regimen for patients undergoing allogeneic haematopoietic stem cell transplant.	38–58 (49)	De novo AML (10) Secondary AML (2)	[47]	Plerixafor administration was safe and well tolerated. Transient AEs possibly related to plerixafor included nausea, dizziness and fatigue. Study not powered to detect benefit of plerixafor on post-allogeneic haematopoietic stem cell transplant relapse rates and survival.
Plerixafor	NCT00943943 (Phase 1)	Plerixafor (240 μg/kg adjusted body weight, subcutaneously) combined with G-CSF and sorafenib dose-escalation.	18–84 (58)	R/R AML with FLT3-ITD mutation (28)	[48]	Plerixafor/G-CSF mobilised leukaemic blasts into the peripheral blood. Grade 3 bone pain attributable to plerixafor and G-CSF. ORR * of 36%.
BL-8040	NCT01838395 (Phase 2a)	Once daily dose of BL-8040 (0.5–2 mg/kg) administered as monotherapy for days 1–2, followed by combination of BL-8040 with cytarabine.	(61)	R/R AML (45)	[53]	BL-8040 induced mobilisation, differentiation and cell death in AML blasts. Treatment was safe and well tolerated. BL-8040 (≥1.0 mg/kg) promoted CR + CRi rate of 38%. BL-8040-cytarabine combination improved the historical response rate achieved with cytarabine alone.
LY2510924	NCT02652871 (Phase 1)	Dose escalation at two doses (10, 20 mg/day), administered as monotherapy daily for 7 days, followed by idarubicin and cytarabine combined with LY2510924.	19–70 (55)	R/R AML (11)	[55]	LY2510924 monotherapy promoted strong mobilisation of leukaemic blasts. Combination of LY2510924 with idarubicin and cytarabine was safe. Common agent-related AEs included diarrhea, nausea/vomiting, mucositis, constipation, and pruritus. ORR * of 36%. Current doses did not completely suppress CXCR4 receptor occupancy.
Ulocuplumab	N/R (Phase 1)	Dose escalation at 0.3, 1, 3, and 10 mg/kg. Cohorts at 0.3 mg/kg received three weekly doses as monotherapy, followed by the same doses with MEC chemotherapy. Cohorts at 1, 3, and 10 mg/kg received 1 weekly dose as monotherapy, followed by the same combination regimen.	N/R	R/R AML (24)	[60]	Ulocuplumab mobilised leukaemic blasts and leukaemic stem cells into the peripheral blood. In some patients, leukaemic blasts and leukaemic stem cells remained in the peripheral circulation for days. Apoptosis induction by antibody was demonstrated by increased Annexin V stained leukaemic blasts/leukaemic stem cells following antibody exposure.
Ulocuplumab	N/R (Phase 1)	Dose escalation (0.3, 1, 3, and 10 mg/kg) was given as a single infusion a week prior to MEC treatment, followed by 3 additional weekly doses per MEC cycle thereafter. For expansion, 10 mg/kg ulocuplumab + MEC regimen was used.	21–79 (58)	R/R AML (73)	[61]	Ulocuplumab mobilised leukaemic blasts into the peripheral blood. Transient and mild/moderate thrombocytopenia was the only AE associated with ulocuplumab. CR/CRi rate of 51%. Ulocuplumab–MEC combination improved the historical response rate achieved with MEC alone.
CX-01	NCT02056782 (Phase 1)	For cytarabine/idarubicin induction, CX-01 (4 mg/kg) was given over 30 min after the first dose of idarubicin, followed by continuous infusion of 0.25 mg/kg per hour thereafter. For consolidation, CX-01 (4 mg/kg) was given over 30 min after the first dose of cytarabine, followed by continuous infusion of 0.25 mg/kg per hour thereafter.	22–74 (54)	De novo AML (11) CMML-2 (1)	[63]	Combining CX-01 treatment with standard AML therapy was safe and well tolerated. No serious AEs related to CX-01 were reported. CR rate of 92%. CX-01 improved treatment efficacy and count recovery in patients with AML.
PF-03084014	A8641014 (Phase 1)	PF-03084014 administered twice weekly (150 mg) in continuous cycles.	18–43 (31)	R/R T-ALL (3) R/R T-LBL (5)	[66]	Most common treatment-related AEs were nausea and vomiting. A patient with T-ALL achieved CR. PF-03084014 treatment displayed anti-leukaemic activity.
BMS-906024	CA216002 (Phase 1)	BMS-906024 was given intravenously weekly at doses of 0.6, 4, and 6 mg.	18–74	R/R T-ALL/T-LBL (25)	[68]	BMS-906024 treatment was relatively well tolerated. Most common AE related to agent was diarrhea. Reduction of (at least 50%) bone marrow blasts in 32%, including one CR and one PR. BMS-906024 showed potential as a Notch-targeting agent in T-ALL.
CWP232291	NCT01398462 (Phase 1)	Dose escalation at 4–334 mg/m^2^. Agent was administered intravenously daily for 7 days every 21 days. MTD was defined at 257 mg/m^2^.	25–81 (64)	R/R AML (64) R/R MDS (5)	[74]	CWP232291 administration was safe and well tolerated. Most common treatment-related AEs were nausea, vomiting, diarrhea, and infusion-related reactions. Low CR (1.56%) and PR (1.56%) rates achieved for patients with AML. Patients with MDS did not respond to treatment. CWP232291 showed minimal/modest efficacy as a single agent.
RG7356/ ARH460-16-2	NCT01641250 (Phase 1)	RG7356 was administered intravenously at dosages ≤2400 mg every other week, or ≤1200 mg weekly or twice weekly.	20–82 (69)	R/R AML (37) TN AML (7)	[77]	Treatment was safe and well tolerated. Majority of treatment-related AEs were transient and mild/moderate, with infusion related reactions being the most frequently observed. Low response rates with 1 CRp and 1 PR. Limited clinical activity with the use of RG7356 as monotherapy.
Uproleselan (GMI-1271)	N/R (Phase 1/2)	Dose escalation at 5–20 mg/kg in combination with MEC in patients with R/R AML. In phase 2, patients were given uproleselan with chemotherapy. RP2D was 10 mg/kg.	R/R patients 26–84 (59) TN patients 60–79 (67)	R/R AML (66) TN AML (25)	[82]	Uproleselan was well tolerated with no increase in AEs, other than low rates of mucositis. CR/CRi rates of 41% at RP2D for R/R patients. CR/CRi rates of 72% for all TN patients. MRD-remission rates of 69% for evaluable R/R AML patients and 56% for evaluable TN AML patients. E-selectin ligand is a predictor of response by uproleselan in R/R AML.
Everolimus	NCT00968253 (Phase 1/2)	Everolimus was given continuously at 5 or 10 mg/day with HyperCVAD treatment. MTD was defined at 5 mg/day.	11–64 (25)	R/R Pre-B ALL (13) R/R T-ALL (10) MPAL (1)	[84]	Combination of everolimus and HyperCVAD was well tolerated with no increase in toxicity detected. Mucositis was defined as the dose limiting toxicity. ORR * of 33% and PR of 8%. Combined CR and PR of 50% and median overall survival of 23 weeks for heavily pre-treated patients with T-ALL.
Everolimus	NCT01523977 (Phase 1b)	Dose escalation at 2, 3, and 5 mg/m^2^/day for 32 days, co-administered with multi-agent reinduction chemotherapy.	2.4–22.8 (11)	Relapsed B-ALL (21) Relapsed T-ALL (1)	[85]	Everolimus combined with reinduction chemotherapy was feasible in childhood ALL with first marrow relapse. Hyperbilirubinemia, neutropenia, transaminitis, hypophosphatemia, and infections identified as potential dose-related AEs. CR2 rate of 86% achieved for all patients. Of those with CR2, 68% achieved a low end-reinduction MRD. Study was not powered to determine efficacy.
Cabozantinib	NCT01961765 (Phase 1)	Dose escalation at 40, 60, and 80 mg daily in 28-day cycles. MTD was defined at 40 mg daily.	27–85 (68)	R/R AML (16) Newly diagnosed AML (2)	[87]	Cabozantinib was well tolerated. Nausea and transaminitis possibly associated with cabozantinib. Treatment reduced circulating blasts in some patients, but none had a marrow response.
Bortezomib	NCT01371981 (Phase 3)	Bortezomib (1.3 mg/m^2^) was incorporated into chemotherapy. Patients were randomly assigned to either standard AML therapy or standard therapy with bortezomib.	0–29.5 (9.2)	De novo AML (1231)	[91]	Bortezomib increased toxicity without improving survival. Peripheral neuropathy and paediatric intensive care unit admissions increased in patients receiving bortezomib. Results do not support addition of bortezomib to standard chemotherapy for de novo AML.
Carfilzomib	NCT01137747 (Phase 1)	Dose escalation at 36, 45, and 56 mg/m^2^, administered as a 30 min infusion on a 28-day cycle.	32–78 (70)	R/R AML (17) R/R ALL (1)	[94]	Carfilzomib was safe and well tolerated amongst participants. No dose limiting toxicity or evidence of AEs related to treatment was observed. 2 PR and 4 SD.
Carfilzomib	NCT02303821 (Phase 1b)	Dose escalation at 27–56 mg/m^2^ for patients treated with VXLD. Patients received one 4-week cycle of induction chemotherapy with VXLD, plus carfilzomib administered intravenously.	1–19 (11)	R/R B-ALL (7) R/R T-ALL (8)	[95]	Carfilzomib in combination with VXLD was tolerable amongst participants. Common AE include sepsis, pancreatitis, and posterior reversible encephalopathy syndrome. ORR * of 67% at the end of consolidation. Efficacy of carfilzomib is promising in this study cohort.
Ixazomib	NCT02070458 (Phase 1/2)	Dose escalation at 1, 2, and 3 mg in combination with MEC treatment. MTD was defined at 1 mg.	31–70 (58)	R/R AML (30)	[96]	Combining ixazomib with MEC was safe and well tolerated. No increase in toxicity of the MEC regimen when combined with ixazomib was observed. ORR * of 53%. The genes *IFI30* and *RORα* were predictors of response.
PR-104	NCT01037556 (Phase 1/2)	Patients received PR-104 at doses ranging from 1.1–4 g/m^2^. For dose expansion, patients were treated at 3 or 4 g/m^2^.	20–79 (62)	R/R AML (40) R/R ALL (10)	[99]	PR-104 induced significant toxicity. Most common grade 3/4 treatment-related AEs were myelosuppression, febrile neutropenia, infection, and enterocolitis. 32% of patients with AML and 20% of patients with ALL demonstrated CR/CRp + MLFS at 3 or 4 g/m^2^. PR-104 decreased leukaemic cells in the hypoxic bone marrow; however, on-target toxicity to the bone marrow was a therapeutic limitation.
Evofosfamide (TH-302)	NCT01149915 (Phase 1)	Evifosfamide administered daily as a 30–60 min infusion (120–550 mg/m^2^; MTD 460 mg/m^2^), or as a continuous intravenous infusion over 120 h (MTD 330 mg/m^2^), on days 1–5 of 21-day cycles.	23–76 (58)	R/R AML (39) R/R ALL (9) CML (1)	[102]	Mucositis reported as primary dose limiting toxicity. Evofosfamide demonstrated limited activity of 6% ORR * in heavily pre-treated patients with advanced disease. Hypoxia markers HIF-1α and CAIX were significantly reduced by therapy in the leukaemic bone marrow.
CA1P (OXi4503)	NCT02576301 (Phase 1B)	CA1P administered at doses ranging from 3.75 to 12.2 mg/m^2^ in combination with cytarabine. MTD defined at 9.76 mg/m^2^.	26–78 (61)	R/R AML (27) R/R MDS (2)	[104]	CA1P (at MTD) in combination with cytarabine was generally well tolerated. Most common treatment-related AEs included febrile neutropenia, hypertension, thrombocytopenia, and anemia. Median overall survival for patients who achieved CR/CRi (528 days) was significantly longer than the median overall survival for patients who did not achieve a CR/CRi (113 days). ORR * of 19%.

Abbreviations: AE, adverse event/effect; AML, acute myeloid leukaemia; ALL, acute lymphoblastic leukaemia; CMML-2, chronic myelomonocytic leukaemia-2; CR, complete remission; CRi, CR with incomplete haematologic recovery; CRp, CR with incomplete platelet recovery; CR2, second complete remission; MEC, mitoxantrone, etoposide, cytarabine; MDS, myelodysplastic syndrome; MLFS, morphological leukaemia-free state; MPAL, mixed phenotype acute leukaemia; MRD, minimal residual disease; MTD, maximum tolerated dose; N/R, not reported; ORR, overall response rate; PR, partial remission; RP2D, recommended phase 2 dose; R/R, relapsed/refractory; SD; stable disease; T-LBL, T-cell lymphoblastic lymphoma; TN, treatment-naïve; VXLD, vincristine, dexamethasone, PEG-asparaginase, daunorubicin. * indicates ORR was defined differently according to each individual study.

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
