# Peer review of "Therapeutic Targeting of the Leukaemia Microenvironment"

_ijms, 2021, doi:10.3390/ijms22136888_

Round 1
Reviewer 1 Report
The review entitled “Therapeutic Targeting of the Leukemia Microenvironment” provides a detailed and current understanding of the regulatory mechanisms associated with leukemia-bone marrow niche interaction and their association with chemoresistance and relapse. The authors further provided perspective on current bone marrow microenvironment targeted therapies and discuss the strengths and limitations associated with these treatment approaches. The review is timely, very comprehensive, nicely written, and adequately covers the literature in the field. I read the manuscript with a strong interest, and I think that this review will benefit both clinicians and researchers.
Some concerns raised are shown below.
- Adipocytic niche (adipocyte-rich microenvironment, fatty acid metabolism) and immune cell microenvironment (macrophages) also play a significant role in leukemia physiology. The authors should discuss it. Additionally, the efficacy of immune based therapies including CD47 inhibition, CAR-T cell therapy in leukemia management should be described.
- Mitochondria are emerging components in the molecular/metabolic exchange between leukemic cells and bone marrow microenvironment, and provides a survival advantage. The authors should include these points and the related strategies targeting intercellular mitochondrial exchange for the elimination of minimal residual disease.
- The cell-extrinsic BM niche factors regulating clonal hematopoiesis and leukemia progression should be discussed.
Reviewer 2 Report
The paper by Vincent Kuek, Anastasia M Hughes and colleagues is a comprehensive, well-written review that describes current progress in therapeutic targeting of the bone marrow environment for leukemia treatment.
The review provides some background on the structure of normal bone marrow hematopoietic environment with references to several recent reviews. Since the section #2 (the bone marrow niche) is quite small, the recommendation would be to highlight the fact that the structure of normal HSCs niche has been extensively studied and previously intensively reviewed, followed by the combined references to several reviews (Ref6, 7, 8). I also recommend citing the review by Samiksha Wasnik et al "HSC Niche: Regulation of Mobilization and Homing" from the Biology and Engineering of Stem Cell Niches https://works.bepress.com/david-baylink/266/
as it provides a brief overview of major factors involved in HSCs trafficking, discussed in your paper from a clinical standpoint.
In summary, your review is very interesting and important for scientific community. I recommend this review for publication.
